# Nonparametric Bayesian Lomax delegate racing for survival analysis with competing risks

**Quan Zhang**
McCombs School of Business
The University of Texas at Austin
Austin, TX 78712
quan.zhang@mccombs.utexas.edu

**Mingyuan Zhou**
McCombs School of Business
The University of Texas at Austin
Austin, TX 78712
mingyuan.zhou@mccombs.utexas.edu

## Abstract

We propose Lomax delegate racing (LDR) to explicitly model the mechanism of survival under competing risks and to interpret how the covariates accelerate or decelerate the time to event. LDR explains non-monotonic covariate effects by racing a potentially infinite number of sub-risks, and consequently relaxes the ubiquitous proportional-hazards assumption which may be too restrictive. Moreover, LDR is naturally able to model not only censoring, but also missing event times or event types. For inference, we develop a Gibbs sampler under data augmentation for moderately sized data, along with a stochastic gradient descent maximum a posteriori inference algorithm for big data applications. Illustrative experiments are provided on both synthetic and real datasets, and comparison with various benchmark algorithms for survival analysis with competing risks demonstrates distinguished performance of LDR.

## 1 Introduction

In survival analysis, one can often use nonparametric approaches to flexibly estimate the survival function from lifetime data, such as the Kaplan–Meier estimator [1], or to estimate the intensity of a point process for event arrivals, such as the isotonic Hawkes process [2] and neural Hawkes process [3] that can be applied to the analysis of recurring events. When exploring the relationship between the covariates and time to events, existing survival analysis methods often parameterize the hazard function with a weighted linear combination of covariates. One of the most popular ones is the Cox proportional hazards model [4], which is semi-parametric in that it assumes a non-parametric baseline hazard rate to capture the time effect. These methods are often applied to population-level studies that try to unveil the relationship between the risk factors and hazard function, such as to what degree a unit increase in a covariate is multiplicative to the hazard rate. However, the interpretability is often obtained by sacrificing model flexibility, because the proportional-hazards assumption is violated when the covariate effects are non-monotonic. For example, both very high and very low ambient temperature were related to high mortality rates in Valencia, Spain, 1991-1993 [5], and a significantly increased mortality rate is associated with both underweight and obesity [6].

To accommodate nonlinear covariate effects such as non-monotonicity, existing (semi-)parametric models often expand the design matrix with transformed data, like the basis functions of smoothing splines [7, 8] and other transformations guided by subjective knowledge. Instead of using hand-designed data transformations, there are several recent studies in machine learning that model complex covariate dependence with flexible functions, such as deep exponential families [9], neural networks [10–12] and Gaussian processes [13]. With enhanced flexibilities, these recent approaches are often good at assessing individual risks, such as predicting a patient's hazard function or survival time. However, except for the Gaussian process whose results are not too difficult to interpret for

low-dimensional covariates, they often have difficulty in explaining how the survival is impacted by which covariates, limiting their use in survive analysis where interpretability plays a critical role. Some approaches discretize the real-valued survival time and model the surviving on discrete time points or intervals [14–17]. They transform the time-to-event modeling problem into regression, classification, or ranking ones, at the expense of losing continuity information implied by the survival time and potentially having inconsistent categories between training and testing.

In survival analysis, it is very common to have competing risks, in which scenario the occurrence of an event under a risk precludes events under any other risks. For example, if the event of interest is death, then all possible causes of death are competing risks to each other, since a subject that died of one cause would never die of any other cause. Apart from modeling the time to event, in the presence of competing risks, it is also important to model the event type, or under which risk the event is likely to occur first. Though one can censor subjects with an occurrence of the event under a competing risk other than the risk of special interest, so that every survival model that can handle censoring is able to model competing risks, it is problematic to violate the principle of non-informative censoring [18, 19]. The analysis of competing risks should be carefully designed and people often model two types of hazard functions, cause specific [20, 21] and subdistribution [20–22] hazard functions. The former applies to studying etiology of diseases, while the latter is favorable when developing prediction models and risk-censoring systems [19].

In the analysis of competing risks, there is also a trade-off between interpretability and flexibility. The aforementioned cause specific and subdistribution hazard functions use a Cox model with competing risk [19, 23] and a Fine-Gray subdistribution model [22], respectively, which are both proportional hazard models. Both models are semi-parametric, and assume that the hazard rate is proportional to the exponential of the inner product of the covariate and regression coefficient vectors, along with a nonparametric baseline hazard function. However, the existence of non-monotonic covariate effects can easily challenge and break the proportional-hazards assumption inherited from their corresponding single-risk model. This barrier has been surmounted by nonparametric approaches, such as random survival forests [24], Gaussian processes with a single layer [25] or two [26], and classification-based neural networks that discretize the survival time [27]. These models are designed for competing risks, using the covariates as input and the survival times (or their monotonic transformation) or probabilities as output. Though having good model fit, the non-parametric approaches are specifically used for studies at an individual level, such as predicting the survival time, but not able to tell how the covariates affect the survival or cumulative incidence functions [22, 28]. Moreover, it might be questionable for Alaa and van der Schaar [26] to assume a normal distribution on survival times which are positive almost surely and asymmetric in general.

To this end, we construct Lomax delegate racing (LDR) survival model, a gamma process based nonparametric Bayesian hierarchical model for survival analysis with competing risks. The LDR survival model utilizes the race of exponential random variables to model both the time to event and event type and subtype, and uses the summation of a potentially countably infinite number of covariate-dependent gamma random variables as the exponential distribution rate parameters. It is amenable to not only censoring data, but also missing event types or event times. Code for reproducible research is available at https://github.com/zhangquan-ut/Lomax-delegate-racing-for-survival-analysis-with-competing-risks.

## 2 Exponential racing and survival analysis

Let $t \sim \mathrm{Exp}(\lambda)$ represent an exponential distribution, with probability density function (PDF) $f(t \mid \lambda) = \lambda e^{-\lambda t}$, $t \in \mathbb{R}_+$, where $\mathbb{R}_+$ represents the nonnegative side of the real line, and $\lambda > 0$ is the rate parameter such that $\mathbb{E}[t] = \lambda^{-1}$ and $\mathrm{Var}[t] = \lambda^{-2}$. Shown below is a well-known property that characterizes a race among independent exponential random variables [29, 30].

**Property 1** (Exponential racing ). *If $t_j \sim \mathrm{Exp}(\lambda_j)$, where $j = 1, \ldots, J$, are independent to each other, then $t = \min\{t_1, \ldots, t_J\}$ and the argument of the minimum $y = \mathrm{argmin}_{j \in \{1,\ldots,J\}} t_j$ are independent, satisfying*

$$t \sim \mathrm{Exp}\left(\sum_{j=1}^J \lambda_j\right), \; y \sim \mathrm{Categorical}\left(\lambda_1 \Big/ \sum_{j=1}^J \lambda_j, \cdots, \lambda_J \Big/ \sum_{j=1}^J \lambda_j\right). \quad (1)$$

Suppose there is a race among teams $j = 1, \cdots, J$, whose completion times $t_j$ follow $\mathrm{Exp}(\lambda_j)$, with the winner being the team with the minimum completion time. Property 1 shows the winner's

completion time $t$ still follows an exponential distribution and is independent of which team wins the race. In the context of survival analysis, if we consider a competing risk as a team and the latent survival time under this risk as the completion time of the team, then $t$ will be the observed time to event (or failure time) and $y$ the event type (or cause of failure). Exponential racing not only describes a natural mechanism of competing risks, but also provides an attractive modeling framework amenable to Bayesian inference, as conditioning on $\lambda_j$'s, the joint distribution of the event type $y$ and time to event $t$ becomes fully factorized as

$$P(y, t \,|\, \{\lambda_j\}_{1,J}) = \lambda_y e^{-t \sum_{j=1}^{J} \lambda_j}. \tag{2}$$

In survival analysis, it is rarely the case that both $y$ and $t$ are observed for all observations, and one often needs to deal with missing data and right or left censoring. We write $t \sim \mathrm{Exp}_\Psi(\lambda)$ as a truncated exponential random variable defined by PDF $f_\Psi(t \,|\, \lambda) = \lambda e^{-\lambda t} / \int_\Psi \lambda e^{-\lambda u} du$, where $t \in \Psi$ and $\Psi$ is an open interval on $\mathbb{R}_+$ representing censoring. Concretely, $\Psi$ can be $(T_{r.c.}, \infty)$, indicating right censoring with censoring time $T_{r.c.}$, can be $(0, T_{l.c.})$, indicating left censoring with censoring time $T_{l.c.}$, or can be a more general case $(T_1, T_2), T_2 > T_1$.

If we do not observe $y$ or $t$, or there exists censoring, we have the following two scenarios, for both of which it is necessary to introduce appropriate auxiliary variables to achieve fully factorized likelihoods: 1) If we only observe $y$ (or $t$), then we can draw $t$ (or $y$) shown in (1) as an auxiliary variable, leading to the fully factorized likelihood as in (2); 2) If we do not observe $t$ but know $t \in \Psi$ with $P(t \in \Psi \,|\, \{\lambda_j\}_{1,J}) = \int_\Psi (\sum_j \lambda_j) e^{-\sum_j \lambda_j u} du$, then we draw $t \sim \mathrm{Exp}_\Psi(\sum_j \lambda_j)$, resulting in the likelihood

$$P\left(t, t \in \Psi \,\Big|\, \sum_j \lambda_j\right) = f_\Psi\left(t \,\Big|\, \sum_j \lambda_j\right) P\left(t \in \Psi \,\Big|\, \sum_j \lambda_j\right) = \left(\sum_j \lambda_j\right) e^{-t \sum_j \lambda_j}. \tag{3}$$

Together with $y$, which can be drawn by (1) if it is missing, the likelihood $P(y, t, t \in \Psi \,|\, \{\lambda_j\}_{1,J})$ becomes the same as in (2). The procedure of sampling $t$ and/or $y$, generating fully factorized likelihoods under different censoring conditions, plays a crucial role as a data augmentation scheme that will be used for Bayesian inference of the proposed LDR survival model.

In survival analysis with competing risks, one is often interested in modeling the dependence of the event type $y$ and failure time $t$ on covariates $\boldsymbol{x} = (1, x_1, \ldots, x_V)'$. Under the exponential racing framework, one may simply let $\lambda_j = e^{\boldsymbol{x}'\boldsymbol{\beta}_j}$, where $\boldsymbol{\beta}_j = (\beta_{j0}, \ldots, \beta_{jV})'$ is the regression coefficient vector for the $j$th competing risk or event type. However, the hazard rate for the $j$th competing risk, expressed as $\lambda_j = e^{\boldsymbol{x}'\boldsymbol{\beta}_j}$, is restricted to be log-linear in the covariates $\boldsymbol{x}$. This clear restriction motivates us to generalize exponential racing to Lomax racing, which can have a time-varying hazard rate for each competing risk, and further to Lomax delegate racing, which can use the convolution of a potentially countably infinite number of covariate-dependent gamma distributions to model each $\lambda_j$.

## 3 Lomax and Lomax delegate racings

In this section, we generalize exponential racing to Lomax racing, which relates survival analysis with competing risks to a race of conditionally independent Lomax distributed random variables. We further generalize Lomax racing to Lomax delegate racing, which races the winners of conditionally independent Lomax racings. Below we first briefly review Lomax distribution.

Let $\lambda \sim \mathrm{Gamma}(r, 1/b)$ represent a gamma distribution with $\mathbb{E}[\lambda] = r/b$ and $\mathrm{Var}[\lambda] = r/b^2$. Mixing the rate parameter of an exponential distribution with $\lambda \sim \mathrm{Gamma}(r, 1/b)$ leads to a Lomax distribution [31] $t \sim \mathrm{Lomax}(r, b)$, with shape $r > 0$, scale $b > 0$, and PDF

$$f(t \,|\, r, b) = \int_0^\infty \mathrm{Exp}(t; \lambda) \mathrm{Gamma}(\lambda; r, 1/b) d\lambda = r b^r (t + b)^{-(r+1)}, \ \ t \in \mathbb{R}_+.$$

When $r > 1$, we have $\mathbb{E}[t] = b/(r-1)$, and when $r > 2$, we have $\mathrm{Var}[t] = b^2 r/[(r-1)^2(r-2)]$. The Lomax distribution is a heavy-tailed distribution. Its hazard rate and survival function can be expressed as $h(t) = r/(t+b)$ and $S(t) = (t + b^{-1})^{-r}$, respectively.

### 3.1 Covariate-dependent Lomax racing

We generalize covariate-dependent exponential racing by letting

$$t_j \sim \mathrm{Exp}(\lambda_j), \ \lambda_j \sim \mathrm{Gamma}(r, e^{\boldsymbol{x}'\boldsymbol{\beta}_j}).$$

Marginalizing out $\lambda_j$ leads to $t_j \sim \text{Lomax}(r, e^{-\boldsymbol{x}'\boldsymbol{\beta}_j})$. Lomax distribution was initially introduced to study business failures [31] and has since then been widely used to model the time to event in survival analysis [32–35]. Previous research on this distribution [36–38], however, has been mainly focused on point estimation of parameters, without modeling covariate dependence and performing Bayesian inference. We define Lomax racing as follows.

**Definition 1.** *Lomax racing models the time to event $t$ and event type $y$ given covariates $\boldsymbol{x}$ as*

$$t = t_y, \ y = \operatorname{argmin}_{j \in \{1,\ldots,J\}} t_j, \ t_j \sim \text{Lomax}(r, e^{-\boldsymbol{x}'\boldsymbol{\beta}_j}). \tag{4}$$

To explain the notation, suppose a patient has both diabetes ($j = 1$) and cancer ($j = 2$), then $t_1$ will be the patient's latent survival time under diabetes and $t_2$ under cancer. The patient's observed survival time is $\min(t_1, t_2)$. Note Lomax racing can also be considered as an exponential racing model with multiplicative random effects, since $t_j$ in (4) can also be generated as

$$t_j \sim \text{Exp}(\epsilon_j e^{\boldsymbol{x}'\boldsymbol{\beta}_j}), \ \epsilon_j \sim \text{Gamma}(r, 1).$$

There are two clear benefits of Lomax racing over exponential racing. The first benefit is that given $\boldsymbol{x}$ and $\boldsymbol{\beta}_j$, the hazard rate for the $j$th competing risk, expressed as $r/(t_j + e^{-\boldsymbol{x}'\boldsymbol{\beta}_j})$, is no longer a constant as $e^{\boldsymbol{x}'\boldsymbol{\beta}_j}$. The second benefit is that closed-form Gibbs sampling update equations can be derived, as will be described in detail in Section 4 and the Appendix.

For competing risk $j$, we can also express $t_j \sim \text{Exp}(\epsilon_j e^{\boldsymbol{x}'\boldsymbol{\beta}_j})$, $\epsilon_j \sim \text{Gamma}(r, 1)$ as

$$\ln(t_j) = -\boldsymbol{x}'\boldsymbol{\beta}_j + \varepsilon_j, \ \varepsilon_j = \ln(\varepsilon_{j1}/\varepsilon_{j2}), \ \varepsilon_{j1} \sim \text{Exp}(1), \ \varepsilon_{j2} \sim \text{Gamma}(r, 1).$$

Thus Lomax racing regression uses an accelerated failure time model [18] for each of its competing risks. More specifically, with $S_0(t_j) = (t_j + 1)^{-r}$ and $h_0(t_j) = \frac{r}{t_j+1}$, we have

$$S_j(t_j) = (e^{\boldsymbol{x}'\boldsymbol{\beta}_j} t_j + 1)^{-r} = S_0(e^{\boldsymbol{x}'\boldsymbol{\beta}_j} t_j), \ h_j(t_j) = r(t_j + e^{-\boldsymbol{x}'\boldsymbol{\beta}_j})^{-1} = e^{\boldsymbol{x}'\boldsymbol{\beta}_j} h_0(e^{\boldsymbol{x}'\boldsymbol{\beta}_j} t_j), \tag{5}$$

and hence $e^{-\boldsymbol{x}'\boldsymbol{\beta}_j}$ can be considered as the accelerating factor for competing risk $j$. Considering all $J$ risks, we can express survival function $S(t)$ and hazard function $h(t)$ as

$$S(t) = \prod_{j=1}^{J} S_j(t) = \prod_{j=1}^{J} (e^{\boldsymbol{x}'\boldsymbol{\beta}_j} t + 1)^{-r} = \prod_{j=1}^{J} S_0(e^{\boldsymbol{x}'\boldsymbol{\beta}_j} t), \quad h(t) = \frac{-dS(t)/dt}{S(t)} = \sum_{j=1}^{J} \frac{r}{t + e^{-\boldsymbol{x}'\boldsymbol{\beta}_j}}. \tag{6}$$

The nosology of competing risks is often subjected to human knowledge, diagnostic techniques, and patient population. Diseases with the same phenotype, categorized into one competing risk, might have distinct etiology and different impacts on survival, and thus require different therapies. For example, for a patient with both diabetes and cancer, it can be unknown whether the patient has Type 1 or Type 2 diabetes, where Type 1 is ascribed to insufficient production of insulin from pancreas whereas Type 2 arises from the cells' failure in responding properly to insulin [39]. In this regard, it is often necessary for a model to identify sub-risks within a pre-specified competing risk, which may not only improve the fit of survival time, but also help diagnose new disease subtypes. We develop Lomax delegate racing, assuming that a risk consists of several sub-risks, under each of which the latent failure time is accelerated by the exponential of a weighted linear combination of covariates.

## 3.2 Lomax delegate racing

Based on the idea of Lomax racing that an individual's observed failure time is the minimum of latent failure times under competing risks, we further propose *Lomax delegate racing* (LDR), assuming a latent failure time under a competing risk is the minimum of the failure times under a number of sub-risks appertaining to this competing risk. In particular, let us first denote $G_j \sim \Gamma\text{P}(G_{0j}, 1/c_{0j})$ as a gamma process defined on the product space $\mathbb{R}^+ \times \Omega$, where $\mathbb{R}^+ = \{x : x > 0\}$, $G_{0j}$ is a finite and continuous base measure over a complete separable metric space $\Omega$, and $1/c_{0j}$ is a positive scale parameter, such that $G_j(A) \sim \text{Gamma}(G_{0j}(A), 1/c_{0j})$ for each Borel set $A \subset \Omega$. A draw from the gamma process consists of countably infinite non-negatively weighted atoms, expressed as $G_j = \sum_{k=1}^{\infty} r_{jk} \delta_{\boldsymbol{\beta}_{jk}}$. Now we formally define LDR survival model as follows.

**Definition 2** (Lomax delegate racing). *Given a random draw of a gamma process $G_j \sim \Gamma\text{P}(G_{0j}, 1/c_{0j})$, expressed as $G_j = \sum_{k=1}^{\infty} r_{jk} \delta_{\boldsymbol{\beta}_{jk}}$, for each $j \in \{1, \ldots, J\}$, Lomax delegate racing models the time to event $t$ and event type $y$ given covariates $\boldsymbol{x}$ as*

$$t = t_y, \ y = \operatorname*{argmin}_{j \in \{1,\ldots,J\}} t_j, \ t_j = t_{j\kappa_j}, \ \kappa_j = \operatorname*{argmin}_{k \in \{1,\ldots,\infty\}} t_{jk}, \ t_{jk} \sim \text{Lomax}(r_{jk}, e^{-\boldsymbol{x}'\boldsymbol{\beta}_{jk}}). \tag{7}$$

In contrast to specifying a fixed number of competing risks $J$, the gamma process not only admits a race among a potentially infinite number of sub-risks, but also parsimoniously shrinks toward zero the weights of negligible sub-risks [40, 41], so that the non-monotonic covariate effects on the failure time under a competing risk can be interpreted as the *minimum*, which is a nonlinear operation, of failure times under sub-risks whose accelerating factor is log-linear in $x$. As shown in the following Corollary, LDR can also be considered as a generalization of exponential racing, where the exponential rate parameter of each competing risk $j$ is a weighted summation of a countably infinite number of gamma random variables with covariate-dependent weights.

**Corollary 1.** *Lomax delegate racing survival model can also be expressed as*

$$t = t_y, \ y = \operatorname{argmin}_{j \in \{1, \dots, J\}} t_j, \ t_j \sim \operatorname{Exp}\left(\sum_{k=1}^{\infty} e^{x' \beta_{jk}} \tilde{\lambda}_{jk}\right), \ \tilde{\lambda}_{jk} \sim \operatorname{Gamma}(r_{jk}, 1). \quad (8)$$

We provide in the Appendix the marginal distribution of $t$ in LDR for situations where predicting the failure time is of interest. The survival and hazard functions of LDR, which generalize those of Lomax racing in (6), can be expressed as

$$S(t) = \prod_{j=1}^{J} \prod_{k=1}^{\infty} P(T_{jk} > t_j) = \prod_{j=1}^{J} \prod_{k=1}^{\infty} (e^{x' \beta_{jk}} t_j + 1)^{-r_{jk}}, \quad h(t) = \sum_{j=1}^{J} \sum_{k=1}^{\infty} \frac{r_{jk}}{t_j + e^{-x' \beta_{jk}}}. \quad (9)$$

LDR survival model can be considered as a two-phase racing, where in the first phase, for each of the $J$ pre-specified competing risk there is a race among countably infinite sub-risks, and in the second phase, $J$ risk-specific failure times race with each other to eventually determine both the observed failure time $t$ and event type $y$. Moreover, Corollary 1, representing LDR as a single-phase exponential racing, more explicitly explains non-monotonic covariate effects on $t_j$ by writing the exponential rate parameter of $t_j$ as the aggregation of $\{e^{x' \beta_{jk}}\}_{k=1}^{\infty}$ weighted by gamma random variables with the shape parameters as the atom weights of the gamma process $G_j$.

## 4 Bayesian inference

LDR utilizes a gamma process [42] to support countably infinite regression-coefficient vectors for each pre-specified risk. The gamma process $G_j \sim \Gamma\mathrm{P}(G_{0j}, 1/c_{0j})$ has an inherent shrinkage mechanism in that the number of atoms whose weights are larger than a positive constant $\epsilon$ is finite almost surely and follows a Poisson distribution with mean $\int_{\epsilon}^{\infty} r^{-1} e^{-c_{0j} r} dr$. For the convenience of implementation, as in Zhou et al. [43], we truncate the total number of atoms of a gamma process to be $K$ by choosing a finite and discrete base measure as $G_{0j} = \sum_{k=1}^{K} \frac{\gamma_{0j}}{K} \delta_{\beta_{jk}}$. Let us denote $x_i$ and $y_i$ as the covariates and the event type, respectively, for individual $i \in \{1, \dots, n\}$. We express the full hierarchical model of the (truncated) gamma process LDR survival model as

$$y_i = \operatorname*{argmin}_{j \in \{1, \dots, J\}} t_{ij}, \ t_i = \min_j t_{ij}, \ t_{ij} = t_{ij \kappa_{ij}}, \ \kappa_{ij} = \operatorname*{argmin}_{k \in \{1, \dots, K\}} t_{ijk}, \ t_{ijk} \sim \operatorname{Exp}(\lambda_{ijk}),$$

$$\lambda_{ijk} \sim \operatorname{Gamma}(r_{jk}, e^{x'_i \beta_{jk}}), \ r_{jk} \sim \operatorname{Gamma}(\gamma_{0j}/K, 1/c_{0j}), \ \beta_{jk} \sim \prod_{v=0}^{V} \mathcal{N}(0, \alpha_{vjk}^{-1}), \quad (10)$$

where we further let $\alpha_{vjk} \sim \operatorname{Gamma}(a_0, 1/b_0)$. The joint probability given $\{\lambda_{ijk}\}_{jk}$ is

$$P(t_i, \kappa_{iy_i}, y_i \mid \{\lambda_{ijk}\}_{jk}) = P(t_i \mid \{\lambda_{ijk}\}_{jk}) P(\kappa_{iy_i}, y_i \mid \{\lambda_{ijk}\}_{jk}) = \lambda_{iy_i \kappa_{iy_i}} e^{-t_i \sum_{j=1}^{S} \sum_{k=1}^{K} \lambda_{ijk}},$$

which is amenable to posterior simulation for $\lambda_{ijk}$. Let us denote $\operatorname{NB}(x; r, p) = \frac{\Gamma(x+r)}{x! \Gamma(r)} p^x (1-p)^r$ as the likelihood for negative binomial distribution and $\sigma(x) = 1/(1 + e^{-x})$ as the sigmoid function. Further marginalize out $\lambda_{ijk} \sim \operatorname{Gamma}(r_{jk}, e^{x'_i \beta_{jk}})$ leads to a fully factorized joint likelihood as

$$P(t_i, \kappa_{iy_i}, y_i \mid x_i, \{\beta_{jk}\}_{jk}) = t_i^{-1} \prod_j \prod_k \operatorname{NB}\left(\mathbf{1}(\kappa_{iy_i} = k, y_i = j); r_{jk}, \sigma(x'_i \beta_{jk} + \ln t_i)\right), \quad (11)$$

which is amenable to posterior simulation using the data augmentation based inference technique for negative binomial regression [44, 45]. The augmentation schemes of $t_i$ and/or $y_i$ discussed in Section 2 are used to achieve (11) in the presence of censoring or as a remedy for missing data. We describe in detail both Gibbs sampling and maximum a posteriori (MAP) inference in the Appendix.

Table 1: Synthetic data generating process.

| Synthetic data 1 | Synthetic data 2 |
|---|---|
| $t_i = \min(t_{i1}, t_{i2}, 3.5),$ | $t_i = \min(t_{i1}, t_{i2}, 6.5),$ |
| $t_{i1} \sim \mathrm{Exp}(e^{\boldsymbol{x}_i'\boldsymbol{\beta}_1}), t_{i2} \sim \mathrm{Exp}(e^{\boldsymbol{x}_i'\boldsymbol{\beta}_2})$ | $t_{i1} \sim \mathrm{Exp}(1/\cosh(\boldsymbol{x}_i'\boldsymbol{\beta}_1)), t_{i2} \sim \mathrm{Exp}(1/|\sinh(\boldsymbol{x}_i'\boldsymbol{\beta}_2)|)$ |

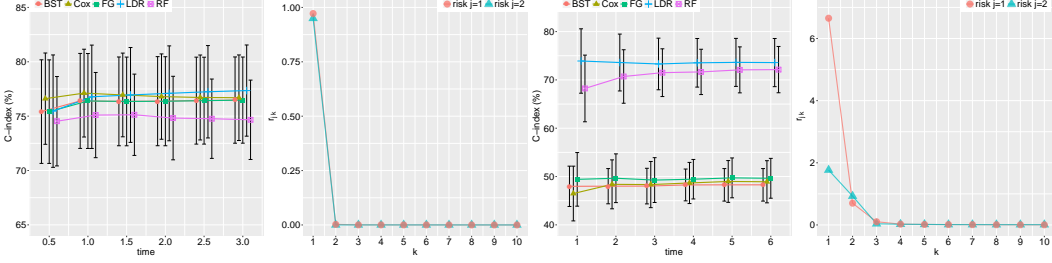

(a) C-index of risk 1 for synthetic data 1.

(b) $r_{jk}$ by descending order for synthetic data 1.

(c) C-index of risk 1 for synthetic data 2.

(d) $r_{jk}$ by descending order for synthetic data 2.

Figure 1: Cause-specific C-indices and shrinkage of $r_{jk}$ by LDR for synthetic data 1 and 2.

## 5 Experimental results

In this section, we validate the proposed LDR model by a variety of experiments using both synthetic and real data. Some data description, implementation of benchmark approaches, and experiment settings are deferred to the Appendix for brevity. In all experiments we exclude from the testing data the observations that have unknown failure times or event types. We compare the proposed LDR survival model, cause-specific Cox proportional hazards model (Cox) [19,23], Fine-Gray proportional subdistribution hazards model (FG) [22] and its boosting algorithm (BST) which is more stable for high-dimensional covariates [46], and random survival forests (RF) [24], which are all designed for survival analysis with competing risks. We show that LDR performs uniformly well regardless of whether the covariate effects are monotonic or not. Moreover, LDR is able to infer the missing cause of death and/or survival time of an observation, both of which in general cannot be handled by these benchmark methods. The model fits of LDR by Bayesian inference via Gibbs sampling and MAP inference via stochastic gradient descent (SGD) are comparable. We will report the results of Gibbs sampling, as it provides an explicit criterion to prune unneeded model capacity (Steps 1 and 8 of Appendix B), avoiding the need of model selection and parameter tuning. For large scale data, performing MAP inference via SGD is recommended if Gibbs sampling takes too long to run a sufficiently large number of iterations. We quantify model performance by cause-specific concordance index (C-index) [23], where the C-index of risk $j$ at time $\tau$ in this paper is computed as

$$\mathcal{C}_j(\tau) = P\left(Score_j(\boldsymbol{x}_i, \tau) > Score_j(\boldsymbol{x}_{i'}, \tau) \,|\, y_i = j \text{ and } [t_i < t_{i'} \text{ or } y_{i'} \neq j]\right),$$

where $i \neq i'$ and $Score_j(\boldsymbol{x}_i, \tau)$ is a prognostic score at time $\tau$ depending on $\boldsymbol{x}_i$ such that its higher value reflects a higher risk of cause $j$. Intuitively, for cause $j$, if patient $i$ died of this cause (i.e., $y_i = j$), and patient $i'$ either died of another cause (i.e., $y_{i'} \neq j$) or died of this cause but lived longer than patient $i$ (i.e., $t_i < t_{i'}$), then it is likely that $Score_j(\boldsymbol{x}_i, \tau)$ for patient $i$ is higher than $Score_j(\boldsymbol{x}_{i'}, \tau)$ for patient $i'$, and the ranking of risks for this pair of patients is concordant. C-index measures such concordance, and a higher value indicates better model performance. Wolbers et al. [23] write C-index as a weighted average of time-dependent AUC that is related to sensitivity, specificity, and ROC curves for competing risks [47]. So a C-index around $0.5$ implies a model failure. A good choice of the prognostic score is the cumulative incidence function, i.e, $Score_j(\boldsymbol{x}_i, \tau) = \mathrm{CIF}_j(i, \tau) = P(t_i \leq \tau, y_i = j)$ [18, 22, 28]. Distinct from a survival function that measures the probability of surviving beyond some time, CIF estimates the probability that an event occurs by a specific time in the presence of competing risks. For LDR given $\{r_{jk}\}$ and $\{\boldsymbol{\beta}_{jk}\}$,

$$\mathrm{CIF}_j(i, \tau) = P(t_i \leq \tau, y_i = j) = \mathbb{E}\left[\frac{\sum_k \lambda_{ijk}}{\sum_{j',k} \lambda_{ij'k}}\left(1 - e^{-\tau \sum_{j',k} \lambda_{ij'k}}\right)\right],$$

where the expectation is taken over $\{\lambda_{jk}\}_{j,k}$, where $\lambda_{ijk} \sim \mathrm{Gamma}(r_{jk}, e^{\boldsymbol{x}_i'\boldsymbol{\beta}_{jk}})$. The expectation can be evaluated by Monte-Carlo estimation if we have a point estimate or a collection of post-burn-in MCMC samples of $r_{jk}$ and $\boldsymbol{\beta}_{jk}$.

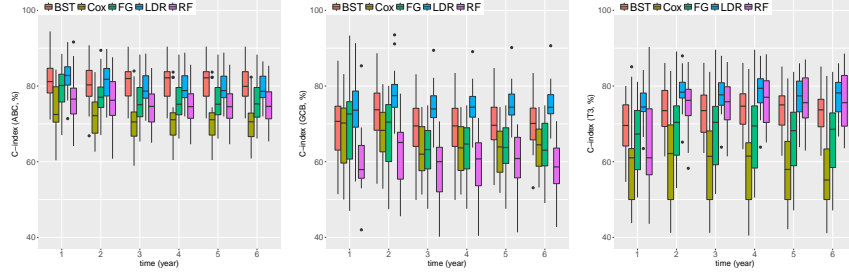

(a) C-index of ABC.　　　　　(b) C-index of GCB.　　　　　(c) C-index of T3.

Figure 2: Cause-specific C-indices for DLBCL data.

## 5.1 Synthetic data analysis

We first simulate two datasets following Table 1, where $\boldsymbol{x}_i \sim \mathrm{N}(\boldsymbol{0}, \mathbf{I}_3)$, to illustrate the unique nonlinear modeling capability of LDR. In Table 1 $t_{ij}$ denotes the latent survival time under risk $j$, $j = 1, 2$ and $t_i$ is the observed time to event. The event type $y_i = \arg\min_j t_{ij}$ if $t_i < T_{r.c.}$, with $y_i = 0$ indicating right censoring if $t_i = T_{r.c.}$, where the censoring time $T_{r.c.} = 3.5$ for data 1 and 6.5 for data 2. We simulate 1,000 random observations, and use 800 for training and the remaining 200 for testing. We randomly take 20 training/testing partitions, on each of which we evaluate the testing cause-specific C-index at time $0.5, 1, 1.5, \cdots, 3$ for data 1 and at time $1, 2, \cdots, 6$ for data 2. The sample mean $\pm$ standard deviation of the estimated cause-specific C-indices of risks 1, and the estimated $r_{jk}$'s of both risks by LDR (from one random training/testing partition but without loss of generality) for data 1 are displayed in panels (a) and (b) of Figure 1, respectively. Analogous plots for data 2 are shown in panels (c) and (d). The testing C-indices of risk 2 are analogous to those of risk 1 for both datasets, thus shown in Figure 5 in the Appendix for brevity.

For data 1 where the survival times under both risks depend on the covariates monotonically, LDR has comparable performance with Cox, FG, and BST, and all these four models slightly outperform RF in terms of the mean values of C-indices. The underperformance of RF in the case of monotonic covariate effects has also been observed in its original paper [24]. For data 2 where the survival time and covariates are not monotonically related, LDR and RF at any time evaluated significantly outperform the other three approaches, all of which fail on this dataset as their C-indices are around 0.5 for both risks. Panels (b) and (d) of Figure 1 show $r_{jk}$ inferred on data 1 and 2, respectively. More specifically, both risks consist of only one sub-risk for data 1. By contrast, two sub-risks of the two respective risks can approximate the complex data generating process of data 2.

## 5.2 Real data analysis

We analyze a microarray gene-expression profile [48] to assess our model performance on real data. The dataset contains a total of 240 patients with diffuse large B-cell lymphoma (DLBCL). Multiple unsuccessful treatments to increase the survival rate suggest that there exist several subtypes of DLBCL that differ in responsiveness to chemotherapy. In the DLBCL dataset, Rosenwald et al. [48] identify three gene-expression subgroups, including activated B-cell-like (ABC), germinal-center B-cell-like (GCB), and type 3 (T3) DLBCL, which may be related to three different diseases as a result of distinct mechanisms of malignant transformation. They also suspect that T3 may be associated with more than one such mechanism. In our analysis, we treat the three subgroups and their potential malignant transformation mechanisms as competing risks from which the patients suffer. As the total number of patients is small which is often the case in survival data, we consider 434 genes that have no missing values across all the patients. Seven of the 434 genes have been reported to be related to clinical phenotypes and four of the seven to have non-monotonic effects on the risk of death [7]. Since some gene expressions may be highly correlated, we follow the same selection procedure of Li and Luan [7] to include as covariates the seven genes, together with another 33 genes having the highest Cox partial score statistic, so that both Cox proportional model and FG subdistribution model for competing risks do not collapse for computational singularity. We use 200 observations for training and the remaining 40 for testing. We take 20 random training/testing partitions and report in Figure 2 boxplots of the testing C-indices evaluated at year $1, 2, \cdots, 6$, by the same five approaches used in the analysis of synthetic datasets.

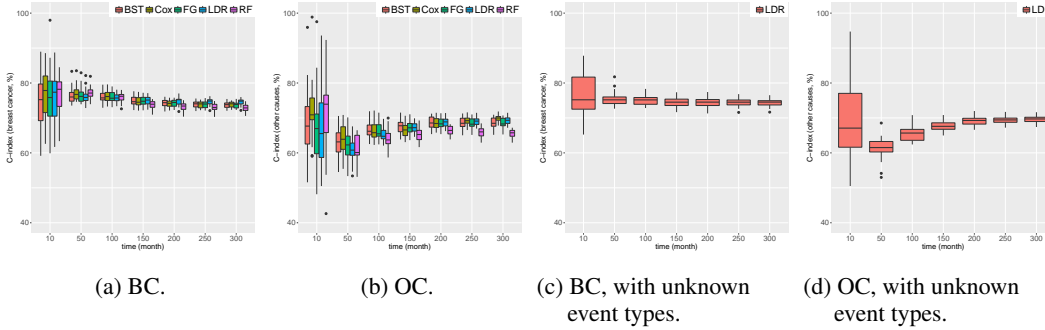

|   (a) BC.   |   (b) OC.   | (c) BC, with unknown event types. | (d) OC, with unknown event types. |

Figure 3: C-indices for SEER breast cancer data.

The boxplots of BST and LDR are roughly comparable for ABC, but the median of LDR is slightly higher than those of BST until year 2, and hereafter slightly lower. For GCB and T3, LDR results in higher median C-indices than all the other benchmarks do at any time evaluated, indicating LDR provides a big improvement in predicting lymphoma CIFs. Interestingly noted is that RF has low performance in both ABC and GCB, but outperforms Cox, FG, and BST and is comparable to LDR in T3. This implies that the gene expressions may have monotonic effects on survival under ABC or GCB, but it is not the case for T3, which can be validated by the fact that LDR learns one sub-risk for ABC and GCB, respectively, and two sub-risks for T3. To better show the improvements of LDR over existing approaches, we calculate the difference of C-indices between LDR and each of the other four benchmarks within each training-testing partition, and report the sample mean and standard deviation across partitions in Table 11 in the Appendix. On average, the improvements of LDR over Cox, FG, and BST are bigger for T3 than those for ABC or GCB, whereas LDR outperforms RF by a larger margin for ABC and GCB than for T3. This shows another advantage of LDR that it fits consistently well regardless of whether the covariate effects are monotonic or not.

We further analyze a publicly accessible dataset from the *Surveillance, Epidemiology, and End Results* (SEER) Program of National Cancer Institute [49]. The SEER dataset we use contains two risks: one is breast cancer and the other is "other causes," which we denote as BC and OC, respectively. It also contains some incomplete observations, each of which with an unknown cause of death but observed uncensored time to death, that can be handled by LDR. The individual covariates include the patients' personal information, such as age, gender, race, and diagnostic and therapy information. More details are deferred to the Appendix.

We first eliminate all observations with unknown causes of death, so we can make comparison between LDR, Cox, FG, BST, and RF. We take 20 random training/testing partitions of the dataset, in each of which $80\%$ of observations are used as training and the remaining $20\%$ as testing. In Figure 3, panels (a) and (b) show the boxplots of C-indices for BC and OC, respectively, obtained from the 20 testing sets by the five models at months $10, 50, 100, \cdots, 300$. For BC the C-indices by LDR are comparable to those by the other four approaches until month 150 and slightly higher afterwards. For the OC the C-indices by LDR are slightly lower than those by Cox, FG, and BST, but become similar after month 100. Also note that RF underperforms the other four approaches since month 100 for BC and month 50 for OC. Comparable C-indices from LDR, Cox, FG, and BST imply monotonic impacts of covariates on survival times under both risks. In fact, for either risk we learn a sub-risk which dominates the others in terms of weights. Furthermore, we analyze the SEER dataset by LDR using the same training/testing partitions, but additionally including the observations having missing causes of death into the 20 training sets, and show the testing C-indices in panels (c) and (d) of Figure 3. We see the testing C-indices are very similar to those in (a) and (b). More importantly, LDR provides a probabilistic inference on missing time to event or missing causes during the model training procedure.

In Appendix E we further provide the Brier scores [50, 51] of each risk in all data sets over time. Brier score quantifies the deviation of predicted CIF's from the actual outcomes and a smaller value implies a better model performance [52]. Tables 2-10 in Appendix E show Brier scores by the models compared on the four data sets, indicating the model out-of-sample prediction performance is basically consistent with those quantified by C-indices. Specifically, for the cases of synthetic data 1, SEER, and both ABC and GCB of DLBCL, where C-indices imply linear covariate effects, the Brier scores are comparable for Cox, FG, BST, and LDR, and slightly smaller than those of RF. For synthetic data 2 and T3 of DLBCL where C-indices imply nonlinear covariate effects, the Brier

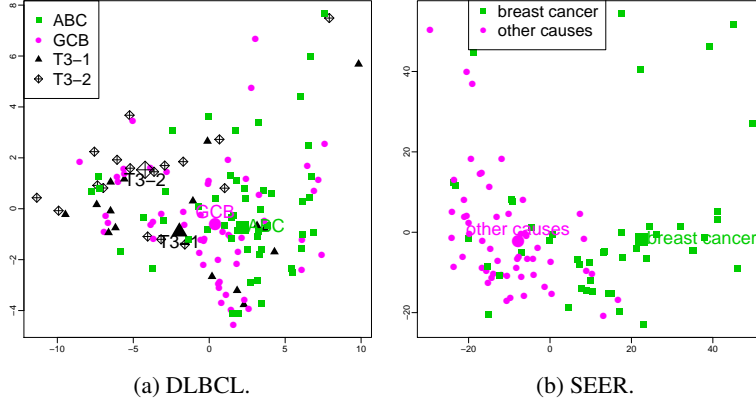

|   (a) DLBCL. | (b) SEER. |
|---|---|

Figure 4: Isomap visualization of the observations and inferred sub-risk representatives.

scores by LDR and RF are smaller than those by Cox, FG, and BST. Moreover, the Brier scores by LDR are slightly larger than those of RF for synthetic data 2 but smaller for T3 of DLBCL.

To show the interpretability of LDR, we visualize representative individuals, each of which suffers from an inferred sub-risk. Specifically, for each inferred sub-risk $k$ under risk $j$, we find the representative by evaluating a weighted average of all uncensored observations as $\sum_i w_{ijk} \boldsymbol{x}_i / \sum_i w_{ijk}$, where $w_{ijk} = \mathbb{E}\left(\frac{\lambda_{ijk}}{\sum_{j'}\sum_{k'}\lambda_{ij'k'}}\right)$, $\lambda_{ijk} \sim \text{Gamma}(\hat{r}_{jk}, e^{\boldsymbol{x}_i'\hat{\boldsymbol{\beta}}_{jk}})$, and $\hat{r}_{jk}$ and $\hat{\boldsymbol{\beta}}_{jk}$ are the estimated values of $r_{jk}$ and $\boldsymbol{\beta}_{jk}$, respectively. The weight $w_{ijk}$ extracts the component of $\boldsymbol{x}_i$ that is likely to make the event of sub-risk $k$ under risk $j$ first occur. Then we implement an Isomap algorithm [53] and visualize in Figure 4 the representatives along with uncensored observations in both DLBCL and SEER. Details are provided in the Appendix.

In Figure 4, small symbols denote uncensored observations and large ones the representatives. Panels (a) and (b) show the representatives suffering from sub-risks in the DLBCL and SEER dataset, respectively. In panel (a), we use green for ABC, pink for GCB, and black for T3. The only representative suffering from ABC (GCB) is surrounded by small green (pink) symbols, indicating they signify a typical gene expression profile that may result in the corresponding malignant transformation. There are two representatives suffering from the two sub-risks of T3, denoted by a large triangle and a large diamond, respectively. They approximately lie in the center of the respective cluster of small triangles and diamonds, which denote patients suffering from the corresponding sub-risks of T3 with an estimated probability greater than $0.5$. The two sub-risks of T3 and the representatives verify the heterogeneity of gene expressions under this risk, and strengthen the belief that T3 consists of more than one type of DLBCL [48]. For the SEER data, we randomly select 100 of the 2088 uncensored observations with known event types for visualization. In panel (b), we use green for BC and pink for OC. LDR learns only one sub-risk for each of these two risks, and place for each risk a representative approximately at the center of the cluster of patients who died of that risk.

# 6 Conclusion

We propose Lomax delegate racing (LDR) for survival analysis with competing risks. LDR models the survival times under risks as a two-phase race of sub-risks, which not only intuitively explains the mechanism of surviving under competing risks, but also helps model non-monotonic covariate effects. We use the gamma process to support a potentially countably infinite number of sub-risks for each risk, and rely on its inherent shrinkage mechanism to remove unneeded model capacity, making LDR be capable of detecting unknown event subtypes without pre-specifying their numbers. LDR admits a hierarchical representation that facilities the derivation of Gibbs sampling under data augmentation, which can be adapted for various practical situations such as missing event times or types. A more scalable (stochastic) gradient descent based maximum a posteriori inference algorithm is also developed for big data applications. Experimental results show that with strong interpretability and outstanding performance, the proposed LDR survival model is an attractive alternative to existing ones for various tasks in survival analysis with competing risks.

**Acknowledgments**

The authors acknowledge the support of Award IIS-1812699 from the U.S. National Science Foundation, and the computational support of Texas Advanced Computing Center.

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
