[Supplementary Material]

# Nonparametric Bayesian Lomax delegate racing for survival analysis with competing risks: Appendix

Quan Zhang and Mingyuan Zhou

## A   Marginal distribution of failure time in LDR

**Theorem 1.** *If $t_i \sim Gamma(1, 1/\lambda_{i\bullet\bullet})$ with $\lambda_{i\bullet\bullet} = \sum_{j,k} \lambda_{ijk}$ and $\lambda_{ijk} \sim Gamma(r_{jk}, 1/b_{ijk})$, the PDF of $t_i$ given $\{r_{jk}\}$ and $\{b_{ijk}\}$ is*

$$f(t_i \,|\, \{r_{jk}\}_{j,k}, \{b_{ijk}\}_{j,k}) = c_i \sum_{m=0}^{\infty} \frac{(\rho_i + m)\delta_{im} b_{i(1)}^{\rho_i + m}}{(t_i + b_{i(1)})^{1 + \rho_i + m}},$$

*and the cumulative density function (CDF) is*

$$P(t_i < q \,|\, \{r_{jk}\}_{j,k}, \{b_{ijk}\}_{j,k}) = 1 - c_i \sum_{m=0}^{\infty} \frac{\delta_{im} b_{i(1)}^{\rho_i + m}}{(q + b_{i(1)})^{\rho_i + m}}, \tag{12}$$

*where $c_i = \prod_{j,k} \left( \frac{b_{ijk}}{b_{i(1)}} \right)^{r_{jk}}$, $b_{i(1)} = \max_{j,k} b_{ijk}$, $\rho_i = \sum_{j,k} r_{jk}$, $\delta_{i0} = 1$, $\delta_{im+1} = \frac{1}{m+1} \sum_{h=1}^{m+1} h\gamma_{ih}\delta_{im+1-h}$ for $m \geq 1$, and $\gamma_{ih} = \sum_{j,k} \frac{r_{jk}}{h} \left( 1 - \frac{b_{ijk}}{b_{i(1)}} \right)^h$.*

It is difficult to utilize the PDF or CDF of $t_i$ in the form of series, but we can use a finite truncation to approximate (12). Concretely, as $P(t_i < \infty \,|\, n_i = 1, \{r_{jk}\}_{j,k}, \{b_{ijk}\}_{j,k}) = c_i \sum_{m=0}^{\infty} \delta_{im} = 1$, we find an $M$ so large that $c_i \sum_{m=0}^{M} \delta_{im}$ close to 1 (say no less than 0.9999), and use $1 - c_i \sum_{m=0}^{M} \frac{\delta_{im} b_{i(1)}^{\rho_i + m}}{(q + b_{i(1)})^{\rho_i + m}}$ as an approximation. Consequently, sampling $t_i$ is feasible by inverting the approximated CDF for general cases. We have tried prediction by finite truncation on some synthetic data and found $M$ is mostly between 10 and 30 which is computationally acceptable.

*Proof.* We first study the distribution of gamma convolution. Specifically, if $\lambda_t \overset{ind}{\sim} Gamma(r_t, 1/b_t)$ with $r_t, b_t \in \mathbb{R}_+$, then the PDF of $\lambda = \sum_{t=1}^{T}$ can be written in a form of series [54] as

$$f(\lambda \,|\, r_1, b_1, \cdots, r_T, b_T) = \begin{cases} c \sum_{m=0}^{\infty} \frac{\delta_m \lambda^{\rho + m - 1} e^{-\lambda b_{(1)}}}{\Gamma(\rho + m)/b_{(1)}^{\rho + m}} & \text{if } \lambda > 0, \\ 0 & \text{otherwise,} \end{cases}$$

where $c = \prod_{t=1}^{T} \left( \frac{b_t}{b_{(1)}} \right)^{r_t}$, $b_{(1)} = \max_t b_t$, $\rho = \sum_{t=1}^{T} r_t$, $\delta_0 = 1$, $\delta_{m+1} = \frac{1}{m+1} \sum_{h=1}^{m+1} h\gamma_h \delta_{m+1-h}$ and $\gamma_h = \sum_{t=1}^{T} r_t \left( 1 - \frac{b_t}{b_{(1)}} \right)^h / h$. [54] proved that $0 < \gamma_{ih} \leq \rho_i b_{i0}^h / h$ and $0 < \delta_{im} \leq \frac{\Gamma(\rho_i + m) b_{i0}^m}{\Gamma(\rho_i) m!}$ where $b_{i0} = max_{j,k}(1 - \frac{b_{ijk}}{b_{i(1)}})$. With $n_i \equiv 1$, we want to show the PDF of $t_i$,

$$
\begin{aligned}
&f(t_i \,|\, \{r_{jk}\}_{j,k}, \{b_{ijk}\}_{j,k}) \\
&= \int_0^{\infty} f(t_i \,|\, \lambda_{i\bullet\bullet}) f(\lambda_{i\bullet\bullet} \,|\, \{r_{jk}\}_{j,k}, \{b_{ijk}\}_{j,k}) d\lambda_{i\bullet\bullet} \\
&= \int_0^{\infty} \sum_{m=0}^{\infty} \frac{c_i \delta_{im} t_i^{n_i - 1} \lambda_{i\bullet\bullet}^{n_i + \rho_i + m - 1} \exp(-t_i \lambda_{i\bullet\bullet} - b_{i(1)} \lambda_{i\bullet\bullet})}{\Gamma(n_i)\Gamma(\rho_i + m)} d\lambda_{i\bullet\bullet} \\
&= \sum_{m=0}^{\infty} \int_0^{\infty} \frac{c_i \delta_{im} t_i^{n_i - 1} \lambda_{i\bullet\bullet}^{n_i + \rho_i + m - 1} \exp(-t_i \lambda_{i\bullet\bullet} - b_{i(1)} \lambda_{i\bullet\bullet})}{\Gamma(n_i)\Gamma(\rho_i + m)} d\lambda_{i\bullet\bullet} \\
&= \frac{c_i t_i^{n_i - 1}}{\Gamma(n_i)} \sum_{m=0}^{\infty} \frac{\Gamma(n_i + \rho_i + m)\delta_{im} b_{i(1)}^{\rho_i + m}}{\Gamma(\rho_i + m)(t_i + b_{i(1)})^{n_i + \rho_i + m}},
\end{aligned} \tag{13}
$$

which suffices to prove the equality in (13). Note that

$$f(t_i \mid n_i, \lambda_{i\bullet\bullet}) f(\lambda_{i\bullet\bullet} \mid \{r_{jk}\}_{j,k}, \{b_{ijk}\}_{j,k})$$

$$=\frac{c_i}{\Gamma(n_i)} t_i^{n_i-1} \lambda_{i\bullet\bullet}^{n_i+\rho_i-1} b_{i(1)}^{\rho_i} \exp(-t_i\lambda_{i\bullet\bullet} - b_{i(1)}\lambda_{i\bullet\bullet}) \sum_{m=0}^{\infty} \frac{\Gamma(\rho_i+m)}{\delta_{im} b_{i(1)}^m \lambda_{i\bullet\bullet}^m}$$

$$\leq\frac{c_i}{\Gamma(n_i)} t_i^{n_i-1} \lambda_{i\bullet\bullet}^{n_i+\rho_i-1} b_{i(1)}^{\rho_i} \exp(-t_i\lambda_{i\bullet\bullet} - b_{i(1)}\lambda_{i\bullet\bullet}) \sum_{m=0}^{\infty} \frac{(b_{i0} b_{i(1)} \lambda_{i\bullet\bullet})^m}{\Gamma(\rho_i)m!}$$

$$=\frac{c_i}{\Gamma(n_i)} t_i^{n_i-1} \lambda_{i\bullet\bullet}^{n_i+\rho_i-1} b_{i(1)}^{\rho_i} \exp(-t_i\lambda_{i\bullet\bullet} - b_{i(1)}\lambda_{i\bullet\bullet} + b_{i0} b_{i(1)} \lambda_{i\bullet\bullet}),$$

which shows the uniform convergence of $f(t_i \mid n_i, \lambda_{i\bullet\bullet}) f(\lambda_{i\bullet\bullet} \mid \{r_{jk}\}_{j,k}, \{b_{ijk}\}_{j,k})$. So the integration and countable summation are interchangeable, and consequently, (13) holds. Next we want to show the CDF of $t_i$,

$$P(t_i < q \mid n_i, \{r_{jk}\}_{j,k}, \{b_{ijk}\}_{j,k}) = \int_0^q \frac{c_i t_i^{n_i-1}}{\Gamma(n_i)} \sum_{m=0}^{\infty} \frac{\Gamma(n_i+\rho_i+m)\delta_{im} b_{i(1)}^{\rho_i+m}}{\Gamma(\rho_i+m)(t_i+b_{i(1)})^{n_i+\rho_i+m}} dt_i$$

$$=\sum_{m=0}^{\infty} \int_0^q \frac{c_i t_i^{n_i-1}}{\Gamma(n_i)} \frac{\Gamma(n_i+\rho_i+m)\delta_{im} b_{i(1)}^{\rho_i+m}}{\Gamma(\rho_i+m)(t_i+b_{i(1)})^{n_i+\rho_i+m}} dt_i. \quad (14)$$

It suffices to show (14). Note that

$$\frac{c_i t_i^{n_i-1}}{\Gamma(n_i)} \sum_{m=0}^{\infty} \frac{\Gamma(n_i+\rho_i+m)\delta_{im} b_{i(1)}^{\rho_i+m}}{\Gamma(\rho_i+m)(t_i+b_{i(1)})^{n_i+\rho_i+m}}$$

$$\leq\frac{c_i t_i^{n_i-1}}{\Gamma(n_i)} \sum_{m=0}^{\infty} \frac{\Gamma(n_i+\rho_i+m) b_{i(1)}^{\rho_i+m}\Gamma(n_i+\rho_i+m)}{\Gamma(\rho_i+m)(t_i+b_{i(1)})^{n_i+\rho_i+m}\Gamma(\rho_i)m!}$$

$$=\frac{c_i t_i^{n_i-1}}{\Gamma(n_i)} \frac{\Gamma(\rho_i+n_i) b_{i(1)}^{\rho_i}}{\Gamma(\rho_i)(t_i+b_{i(1)})^{n_i+\rho_i}} \sum_{m=0}^{\infty} \left[ \frac{\Gamma(n_i+\rho_i+m)}{\Gamma_{n_i+\rho_i}m!} \left( \frac{b_{i(1)}}{t_i+b_{i(1)}} \right)^m \right]$$

$$=\frac{c_i t_i^{n_i-1}\Gamma(\rho_i+n_i) b_{i(1)}^{\rho_i} t_i^{n_i+\rho_i}}{\Gamma(n_i)\Gamma(\rho_i)(t_i+b_{i(1)})^{2(n_i+\rho_i)}}.$$

The last equation holds because the summation of a negative binomial probability mass function is 1. So $f(t_i \mid n_i, \{r_{jk}\}_{j,k}, \{b_{ijk}\}_{j,k})$ is uniformly convergent and (14) holds. Plugging in $n_i = 1$ and calculating the integration, we obtain the CDF of $t_i$. □

## B  Bayesian inference of LDR

With $\boldsymbol{x}_i$ denoting the covariates, $y_i$ event type, and $t_i$ the time to event of observation $i$, we express the full hierarchical form of LDR defined in (7), as

$$t_i = t_{iy_i}, \ y_i = \operatorname*{argmin}_{j\in\{1,\dots,J\}} t_{ij}, \ t_{ij} = t_{ij\kappa_{ij}}, \ \kappa_{ij} = \operatorname*{argmin}_{k\in\{0,\dots,K\}} t_{ijk},$$

$$t_{ijk} \sim \mathrm{Exp}(\lambda_{ijk}), \ \lambda_{ijk} \sim \mathrm{Gamma}(r_{jk}, e^{\boldsymbol{x}_i'\boldsymbol{\beta}_{jk}}), \ k = 1, \cdots, K,$$

$$\boldsymbol{\beta}_{jk} \sim \prod_{g=1}^{P} \mathcal{N}(0, \alpha_{gjk}^{-1}), \ \alpha_{gjk} \sim \mathrm{Gamma}(a_0, 1/b_0), \ r_{jk} \sim \mathrm{Gamma}(\gamma_{0j}/K, 1/c_{0j}),$$

where $k = 1, \cdots, K$, $i = 1, \cdots, n$, and $j = 1, \cdots, J$. We further let $\gamma_{0j} \sim \mathrm{Gamma}(e_0, 1/f_0)$, $c_{0j} \sim \mathrm{Gamma}(e_1, 1/f_1)$, $r_0 \sim \mathrm{Gamma}(e_0, 1/f_0)$, and set $e_0 = f_0 = e_1 = f_1 = 0.01$. Let us denote $T_i$ and $T_{ic}$ as the observed failure time and right censoring time, respectively, for observation $i$. Since left censoring is uncommon and not shown in the real datasets analyzed, we only consider right censoring in our inference and leave to readers other types of censoring which can be analogously done. A Gibbs sampler accommodating missing event times or missing event types proceeds by iterating the following steps.

1. If $y_i$ is observed, we first sample $\kappa_{iy_i}$ by

$$P(\kappa_{iy_i} = k \,|\, y_i, \cdots) = \frac{\lambda_{iy_i k}}{\sum_{k'=1}^{K} \lambda_{iy_i k'}}.$$

   If $y_i$ is unobserved which means a missing event type, we sample $(y_i, \kappa_{iy_i})$ by

$$P(y_i = j, \kappa_{iy_i} = k \,|\, \cdots) = \frac{\lambda_{ijk}}{\sum_{j'=1}^{S} \sum_{k'=1}^{K} \lambda_{ij'k'}}.$$

   We then denote $m_{jk} = \sum_{i:y_i=j} \mathbf{1}(\kappa_{iy_i} = k)$. Define $n_{ijk} = 1$ if $y_i = j$ and $\kappa_{iy_i} = k$, and otherwise $n_{ijk} = 0$. The above sampling procedure means that given the event type $y_i$, we sample the index of the sub-risk that has the minimum survival time.

2. Update $t_i$ for $i = 1, \cdots, n$, $j = 1, \cdots, J$ and $k = 1, \cdots, K$.

   (a) If the failure time $T_i$ is observed, we set $t_i = T_i$.

   (b) Otherwise, we let $t_i = T_{ic} + \tilde{t}_i$, where $(\tilde{t}_i \,|\, -) \sim \mathrm{Exp}(\sum_{j=1}^{S} \sum_{k=1}^{K} \lambda_{ijk})$ and $T_{ic}$ is the right censoring. Note $T_{ic} = 0$ if both event time and censoring time are missing for observation $i$.

3. Sample $(\lambda_{ijk} \,|\, -) \sim \mathrm{Gamma}\left(r_{jk} + n_{ijk}, \frac{e^{\boldsymbol{x}_i' \boldsymbol{\beta}_{jk}}}{1 + t_i e^{\boldsymbol{x}_i' \boldsymbol{\beta}_{jk}}}\right)$, for $i = 1, \cdots, n$, $j = 1, \cdots, J$ and $k = 1, \cdots, K$.

4. Sample $\boldsymbol{\beta}_{jk}$, for $j = 1, \cdots, J$ and $k = 1, \cdots, K$, by Pólya Gamma (PG) data augmentation. First Sample $(\omega_{ijk} \,|\, -) \sim \mathrm{PG}(r_{jk} + n_{ijk}, \boldsymbol{x}_i' \boldsymbol{\beta}_{jk} + \log t_i)$. Then sample $(\boldsymbol{\beta}_{jk} \,|\, -) \sim \mathrm{MVN}(\boldsymbol{\mu}_{jk}, \boldsymbol{\Sigma}_{jk})$ where $\boldsymbol{\Sigma}_{jk} = (V_{jk} + \boldsymbol{X}' \Omega_{jk} \boldsymbol{X})^{-1}$, $\boldsymbol{X} = [\boldsymbol{x}_1', \cdots, \boldsymbol{x}_N']'$, $\Omega_{jk} = \mathrm{diag}(\omega_{1jk}, \cdots, \omega_{njk})$ and $\boldsymbol{\mu}_{jk} = \boldsymbol{\Sigma}_{jk}\left[-\sum_{i=1}^{N}\left(\omega_{ijk} \log t_i + \frac{r_{jk} - n_{ijk}}{2}\right) \boldsymbol{x}_i\right]$. Note to sample from the Pólya-Gamma distribution, we use a fast and accurate approximate sampler of Zhou [41] that matches the first two moments of the original distribution; we set the truncation level of that sampler as five.

5. Sample $(\alpha_{vjk} \,|\, -) \sim \mathrm{Gamma}\left(a_0 + 0.5, 1/(b_0 + 0.5 \beta_{vjk}^2)\right)$ for $v = 0, \cdots, V$, $j = 1, \cdots, J$ and $k = 1, \cdots, K$.

6. Sample $r_{jk}$ and $\gamma_{0j}$, for $j = 1, \cdots, J$ and $k = 1, \cdots, K$, by Chinese restaurant table (CRT) data augmentation [43].

   First sample $(n_{ijk}^{(2)} \,|\, -) \sim \mathrm{CRT}(n_{ijk}, r_{jk})$, and $(l_{jk} \,|\, -) \sim \mathrm{CRT}(\sum_{i=1}^{N} n_{ijk}^{(2)}, \gamma_{0j}/K)$. Then sample $(r_{jk} \,|\, -) \sim \mathrm{Gamma}\left(\sum_{i=1}^{N} n_{ijk}^{(2)} + \gamma_{0j}/K, \frac{1}{c_{0j} + \sum_{i=1}^{N} \log(1 + t_i e^{\boldsymbol{x}_i' \boldsymbol{\beta}_{jk}})}\right)$, and $(\gamma_{0j} \,|\, -) \sim \mathrm{Gamma}\left(e_0 + \sum_{k=1}^{K} l_{jk}, \frac{1}{f_0 - \frac{1}{K}\sum_{k=1}^{K} \log(1 - p_{jk})}\right)$, where $p_{jk} = \frac{\sum_{i=1}^{N} \log(1 + t_i e^{\boldsymbol{x}_i' \boldsymbol{\beta}_{jk}})}{c_{0j} + \sum_{i=1}^{N} \log(1 + t_i e^{\boldsymbol{x}_i' \boldsymbol{\beta}_{jk}})}$.

7. Sample $(c_{0j} \,|\, -) \sim \mathrm{Gamma}\left(e_1 + \gamma_{0j}, \frac{1}{f_1 + \sum_{k=1}^{K} r_{jk}}\right)$ for $j = 1, \cdots, J$.

8. For $j = 1, \cdots, J$ and $k = 1, \cdots, K$, prune sub-risk $k$ of risk $j$ for all observations if $m_{jk} = 0$, by setting $\lambda_{ijk} \equiv 0$ and $t_{ijk} \equiv \infty$ for $\forall i$.

## C  Maximum a posteriori estimation

With the reparameterization that $\lambda_{ijk} = \tilde{\lambda}_{ijk} e^{\boldsymbol{x}_i' \boldsymbol{\beta}_{jk}}$ where $\tilde{\lambda}_{ijk} \overset{iid}{\sim} \mathrm{Gamma}(r_{jk}, 1)$ we first find $p_i$, the likelihood of observation $i$ having event type $y_i$ at event time $t_i$.

$$p_i = \mathbb{E}\left(P(t_i, y_i \,|\, \boldsymbol{\lambda}_i)\right) \equiv \int (p_{t_i} \times p_{y_i}) \, p(\tilde{\boldsymbol{\lambda}}_i \,|\, \boldsymbol{r}) d\tilde{\boldsymbol{\lambda}}_i$$

where $\tilde{\boldsymbol{\lambda}}_i = \{\tilde{\lambda}_{ijk}\}_{j,k}$, $p(\tilde{\boldsymbol{\lambda}}_i \,|\, \boldsymbol{r}) = \prod_{j,k} \mathrm{Gamma}(r_{jk}, 1)$, $\boldsymbol{r} = \{r_{jk}\}_{j,k}$, $\mathrm{Gamma}(r_{jk}, 1)$ is the pdf of a gamma distribution with shape $r_{jk}$ and scale 1, and

$$p_{t_i} = \begin{cases} (\sum_{j,k} \tilde{\lambda}_{ijk} e^{x_i' \beta_{jk}}) \exp\left\{-t_i \sum_{jk} \tilde{\lambda}_{ijk} e^{x_i' \beta_{jk}}\right\} & \text{if } t_i \text{ is uncensored and observed,} \\ \exp\left\{-T_{ic} \sum_{jk} \tilde{\lambda}_{ijk} e^{x_i' \beta_{jk}}\right\} & \text{if } t_i \text{ is right censored at } T_{ic}, \text{ i.e., } t_i > T_{ic}, \\ 1 & \text{if } t_i \text{ is missing, but } y_i \text{ is not,} \end{cases}$$

$$p_{y_i} = \begin{cases} \dfrac{\sum_k \tilde{\lambda}_{iy_ik} e^{x_i' \beta_{y_ik}}}{\sum_{j,k} \tilde{\lambda}_{ijk} e^{x_i' \beta_{jk}}} & \text{if } y_i \text{ is not missing,} \\ 1 & \text{if } y_i \text{ is missing, but } t_i \text{ is not.} \end{cases}$$

Note that we do not define $P(t_i, y_i \,|\, \boldsymbol{\lambda}_i)$ if both $t_i$ and $y_i$ are missing and remove such observations from data. We write $p_{t_i} \equiv p_t(\tilde{\boldsymbol{\lambda}}_i \,|\, \boldsymbol{r})$ and $p_{y_i} \equiv p_y(\tilde{\boldsymbol{\lambda}}_i \,|\, \boldsymbol{r})$.

Imposing a prior $p(\boldsymbol{\beta}_{jk})$ on $\boldsymbol{\beta}_{jk}$ and $p(r_{jk})$ on $r_{jk}$, the log posterior is

$$\log P = \sum_i \log p_i + \sum_{j,k} \log p(\boldsymbol{\beta}_{jk}) + \sum_{j,k} \log p(r_{jk}) + C \tag{15}$$

where $C$ is a constant function of $\{\boldsymbol{\beta}_{jk}\}$ and $\{r_{jk}\}$. In practice we assume a Student's $t$ distribution with degrees of freedom $a$ on each element of $\boldsymbol{\beta}_{jk}$ and a Gamma$(0.01/K, 1/0.01)$ prior on $r_{jk}$. We also found a Gamma$(1/K, 1)$ prior on $r_{jk}$ or an $l^2$-regularizer, $0.001||\boldsymbol{r}||_2$, is more numerically stable. Then we have

$$\log P = \sum_i \log p_i + \sum_{v,j,k} -\frac{a+1}{2} \log\left(1 + \beta_{vjk}^2/a\right) + \sum_{j,k} \left[(0.01/K - 1)\log r_{jk} - 0.01 r_{jk}\right] + c$$

where $c$ is also a constant function of $\{\boldsymbol{\beta}_{jk}\}$ and $\{r_{jk}\}$. For simplicity, we define $\boldsymbol{\beta} = \{\boldsymbol{\beta}_{jk}\}_{j,k}$. We want to maximize $\log P$ with respect to $\boldsymbol{\beta}$ and $\boldsymbol{r}$. The difficulty lies in $p_i$ being the expectation of $p_{t_i} \times p_{y_i}$ over $\tilde{\boldsymbol{\lambda}}_i$ which is a random variable parameterized by $\boldsymbol{r}$. Now we show how to approximate the derivatives of $\log p_i$ by Monte-Carlo simulation and score function gradients. Specifically,

$$\nabla_{\boldsymbol{\beta}} \log p_i = \frac{\int [\nabla_{\boldsymbol{\beta}} (p_{t_i} \times p_{y_i})] \, p(\tilde{\boldsymbol{\lambda}}_i \,|\, \boldsymbol{r}) d\tilde{\boldsymbol{\lambda}}_i}{\int (p_{t_i} \times p_{y_i}) \, p(\tilde{\boldsymbol{\lambda}}_i \,|\, \boldsymbol{r}) d\tilde{\boldsymbol{\lambda}}_i} \approx \frac{\frac{1}{M} \sum_{m=1}^M \nabla_{\boldsymbol{\beta}} \left[p_t(\tilde{\boldsymbol{\lambda}}_i^{(m)} \,|\, \boldsymbol{r}) \times p_y(\tilde{\boldsymbol{\lambda}}_i^{(m)} \,|\, \boldsymbol{r})\right]}{\frac{1}{M} \sum_{m=1}^M \left[p_t(\tilde{\boldsymbol{\lambda}}_i^{(m)} \,|\, \boldsymbol{r}) \times p_y(\tilde{\boldsymbol{\lambda}}_i^{(m)} \,|\, \boldsymbol{r})\right]} \tag{16}$$

where $M$ is a reasonably large number, say 10, $\tilde{\boldsymbol{\lambda}}_i^{(m)} = \{\tilde{\lambda}_{ijk}^{(m)}\}_{jk}$ and $\tilde{\lambda}_{ijk}^{(m)} \overset{iid}{\sim} \text{Gamma}(r_{jk}, 1)$, $\forall i = 1, \cdots, n$ and $m = 1, \cdots, M$. With the fact that $\nabla_{\boldsymbol{r}} p(\tilde{\boldsymbol{\lambda}}_i \,|\, \boldsymbol{r}) = p(\tilde{\boldsymbol{\lambda}}_i \,|\, \boldsymbol{r}) \nabla_{\boldsymbol{r}} \log p(\tilde{\boldsymbol{\lambda}}_i \,|\, \boldsymbol{r})$,

$$\begin{aligned} \nabla_{\boldsymbol{r}} \log p_i &= \frac{\int \nabla_{\boldsymbol{r}} \left[(p_{t_i} \times p_{y_i}) \, p(\tilde{\boldsymbol{\lambda}}_i \,|\, \boldsymbol{r})\right] d\tilde{\boldsymbol{\lambda}}_i}{\int (p_{t_i} \times p_{y_i}) \, p(\tilde{\boldsymbol{\lambda}}_i \,|\, \boldsymbol{r}) d\tilde{\boldsymbol{\lambda}}_i} \\ &= \frac{\int (p_{t_i} \times p_{y_i}) \nabla_{\boldsymbol{r}} \log p(\tilde{\boldsymbol{\lambda}}_i \,|\, \boldsymbol{r}) p(\tilde{\boldsymbol{\lambda}}_i \,|\, \boldsymbol{r}) d\tilde{\boldsymbol{\lambda}}_i}{\int (p_{t_i} \times p_{y_i}) \, p(\tilde{\boldsymbol{\lambda}}_i \,|\, \boldsymbol{r}) d\tilde{\boldsymbol{\lambda}}_i} \\ &\approx \frac{\frac{1}{M} \sum_{m=1}^M p_t(\tilde{\boldsymbol{\lambda}}_i^{(m)} \,|\, \boldsymbol{r}) \times p_y(\tilde{\boldsymbol{\lambda}}_i^{(m)} \,|\, \boldsymbol{r}) \nabla_{\boldsymbol{r}} \log p(\tilde{\boldsymbol{\lambda}}_i^{(m)} \,|\, \boldsymbol{r})}{\frac{1}{M} \sum_{m=1}^M \left[p_t(\tilde{\boldsymbol{\lambda}}_i^{(m)} \,|\, \boldsymbol{r}) \times p_y(\tilde{\boldsymbol{\lambda}}_i^{(m)} \,|\, \boldsymbol{r})\right]} \\ &= \sum_{m=1}^M \frac{p_t(\tilde{\boldsymbol{\lambda}}_i^{(m)} \,|\, \boldsymbol{r}) \times p_y(\tilde{\boldsymbol{\lambda}}_i^{(m)} \,|\, \boldsymbol{r})}{\sum_{m'=1}^M \left[p_t(\tilde{\boldsymbol{\lambda}}_i^{(m')} \,|\, \boldsymbol{r}) \times p_y(\tilde{\boldsymbol{\lambda}}_i^{(m')} \,|\, \boldsymbol{r})\right]} \nabla_{\boldsymbol{r}} \log p(\tilde{\boldsymbol{\lambda}}_i^{(m)} \,|\, \boldsymbol{r}). \end{aligned} \tag{17}$$

Therefore, we can approximate the derivatives of $\log P$ with respect to $\boldsymbol{\beta}$ and $\boldsymbol{r}$ by plugging in (16) and (17), respectively, and maximize $-\log P$ by (stochastic) gradient descent.

# D    Description of SEER data and experiment settings

## D.1    SEER data for survival analysis

We use breast cancer data from Surveillance, Epidemiology, and End Results Program (SEER) of National Cancer Institute between 1973 and 2003. There are two causes of death; the first is breast

cancer and the second is *other causes* treated as a whole. Explanatory variables include age of diagnosis, gender, race, marital status, historic stage, behavior type, tumor size, tumor extension, number of malignant tumors, number of regional nodes containing tumor, number of regional nodes that are examined or removed, confirmation type and surgery type. We use dummies for all categorical variables and select a subset of patient collected from the hospital *C503* so that we do not have to consider site effects. We exclude observations with any missing values in explanatory variables. Finally, there are 2647 and 4166 observations in our data if we exclude and include observations with a missing cause of death, respectively.

## D.2 Experiment settings

We run $10,000$ interations of Gibbs sampler for LDR with the gamma process truncated at $K = 10$ for all experiments, take the first $8,000$ as burn-in, and estimate CIF by averaging its estimators from the last $2,000$ iterations. For random survial forests, we set the number of trees equal to $100$ and the number of splits equal to $2$ as suggested by Ishwaran et al. [24]. We use R for all the analysis: C-indices are estimated by package pec [55], the Cox model by `riskRegression` [56], FG by `cmprsk` [57], BST by `CoxBoost` [58], and RF by `randomForestSRC` [59].

Isomap algorithm is often used for nonlinear dimensionality reduction. We first find five nearest neighbors of each observation, and then construct a neighborhood graph where an observation is connected to another with the edge length equal to the Euclidean distance if it is a 5-nearest neighbor. We calculate the shortest path between two nodes of the graph by Floyd–Warshall algorithm [60] and obtain a geodesic distance matrix with which we compute two-dimensional embeddings by classical multidimensional scaling [61].

## E  Additional experimental results

We first show in Table 2 through Table 8 the Brier score at the evaluation time for each risk of the synthetic data sets, SEER and DLBCL data, respectively. Brier score (BS) for risk $j$ at time $\tau$ can be estimated by $\text{BS}_j(\tau) = \frac{1}{n} \sum_{i=1}^{n} \left[ \mathbf{1}(t_i \leq \tau, y_i = j) - P(t_i \leq \tau, y_i = j) \right]^2$, with a smaller value indicating a better model fit. Note that the model performance quantified by Brier score is basically consistent with quantified by C-indices. For the cases like synthetic data 1, SEER and ABC and GCB of DLBCL, where covariates are believed to be linearly influential by C-indices, the Brier scores are comparable for Cox, FG, BST and LDR, and slightly smaller than those of RF. For synthetic data 2 and T3 of DLBCL where C-indices imply nonlinear covariate effects, the Brier scores of LDR and RF are smaller than those of Cox, FG and BST. Moreover, the Brier score of LDR is slightly larger than those of RF for synthetic data 2 but smaller for T3 of DLBCL.

Table 2: Brier score for risk 1 of synthetic data 1.

|  | $\tau = 0.5$ | $\tau = 1$ | $\tau = 1.5$ | $\tau = 2$ | $\tau = 2.5$ | $\tau = 3$ |
|---|---|---|---|---|---|---|
| Cox | .165±.012 | **.166**±.010 | .165±.010 | .166±.012 | **.164**±.012 | **.162**±.012 |
| FG | .168±.010 | .167±.010 | .166±.009 | .166±.012 | **.164**±.013 | **.162**±.012 |
| BST | .167±.010 | **.166**±.010 | .166±.010 | .166±.010 | .166±.011 | .165±.010 |
| RF | .173±.013 | .175±.012 | .171±.012 | .172±.014 | .172±.014 | .170±.014 |
| LDR | **.164**±.014 | **.166**±.011 | **.164**±.010 | **.165**±.012 | **.164**±.013 | **.162**±.013 |

Table 3: Brier score for risk 2 of synthetic data 1.

|  | $\tau = 0.5$ | $\tau = 1$ | $\tau = 1.5$ | $\tau = 2$ | $\tau = 2.5$ | $\tau = 3$ |
|---|---|---|---|---|---|---|
| Cox | **.152**±.011 | **.158**±.014 | **.158**±.015 | .157±.015 | **.157**±.014 | .159±.014 |
| FG | .157±.012 | .159±.014 | .159±.015 | .158±.015 | .158±.014 | .159±.014 |
| BST | .158±.013 | **.158**±.013 | **.158**±.013 | .158±.013 | .158±.013 | .158±.013 |
| RF | .164±.012 | .166±.015 | .166±.016 | .164±.015 | .165±.014 | .165±.014 |
| LDR | **.152**±.012 | **.158**±.014 | **.158**±.016 | **.156**±.015 | **.157**±.014 | **.158**±.014 |

We show in Figure 5 the C-indices of risk 2 for synthetic data 1 and 2 used in Section 5.1. The C-indices of risk 2 for data 1 are very similar to those of risk 1 as in panel (a) of Figure 1; LDR, Cox,

Table 4: Brier score for risk 1 of synthetic data 2.

|  | $\tau = 1$ | $\tau = 2$ | $\tau = 3$ | $\tau = 4$ | $\tau = 5$ | $\tau = 6$ |
|---|---|---|---|---|---|---|
| Cox | .206±.008 | .235±.006 | .241±.005 | .242±.005 | .243±.005 | .243±.005 |
| FG | .206±.008 | .235±.006 | .241±.006 | .242±.005 | .243±.005 | .243±.005 |
| BST | .234±.005 | .234±.005 | .234±.005 | .234±.005 | .234±.005 | .234±.005 |
| RF | **.186**±.010 | **.193**±.011 | **.188**±.011 | **.186**±.010 | **.184**±.010 | **.183**±.010 |
| LDR | .193±.007 | .194±.007 | .191±.006 | .191±.006 | .191±.006 | .191±.006 |

Table 5: Brier score for risk 2 of synthetic data 2.

|  | $\tau = 1$ | $\tau = 2$ | $\tau = 3$ | $\tau = 4$ | $\tau = 5$ | $\tau = 6$ |
|---|---|---|---|---|---|---|
| Cox | .251±.002 | .247±.003 | .245±.004 | .244±.004 | .244±.004 | .244±.004 |
| FG | .251±.002 | .247±.003 | .245±.004 | .244±.004 | .244±.004 | .244±.005 |
| BST | .245±.003 | .245±.003 | .245±.003 | .245±.003 | .245±.003 | .245±.003 |
| RF | **.178**±.011 | **.182**±.010 | **.181**±.010 | **.182**±.010 | **.182**±.010 | **.183**±.010 |
| LDR | .204±.006 | .199±.005 | .197±.005 | .198±.005 | .197±.005 | .199±.005 |

Table 6: Brier score for ABC of DLBCL.

|  | $\tau = 1$ | $\tau = 2$ | $\tau = 3$ | $\tau = 4$ | $\tau = 5$ | $\tau = 6$ |
|---|---|---|---|---|---|---|
| Cox | .162±.056 | .190±.055 | .196±.058 | .202±.054 | .196±.053 | .202±.054 |
| FG | .159±.057 | .185±.058 | .198±.058 | .196±.057 | .196±.056 | .199±.055 |
| BST | .136±.045 | .146±.045 | **.163**±.044 | **.154**±.044 | **.150**±.045 | **.152**±**.044** |
| RF | .156±.052 | .173±.055 | .196±.051 | .198±.051 | .198±.051 | .200±.051 |
| LDR | **.131**±.050 | **.143**±.050 | **.163**±.047 | .158±.045 | .155±.043 | .156±.041 |

Table 7: Brier score for GCB of DLBCL.

|  | $\tau = 1$ | $\tau = 2$ | $\tau = 3$ | $\tau = 4$ | $\tau = 5$ | $\tau = 6$ |
|---|---|---|---|---|---|---|
| Cox | .138±.048 | .212±.051 | .266±.061 | 268±.062 | .265±.062 | .277±.063 |
| FG | .137±.046 | .206±.064 | .268±.059 | .265±.062 | .267±.063 | .273±.064 |
| BST | .133±.046 | .204±.056 | .262±.042 | .252±.036 | .253±.048 | .257±.041 |
| RF | .137±.038 | .197±.054 | .248±.050 | .247±.046 | .253±.050 | .262±.053 |
| LDR | **.129**±.035 | **.179**±.052 | **.242**±.053 | **.236**±.050 | **.237**±.052 | **.244**±.052 |

Table 8: Brier score for T3 of DLBCL.

|  | $\tau = 1$ | $\tau = 2$ | $\tau = 3$ | $\tau = 4$ | $\tau = 5$ | $\tau = 6$ |
|---|---|---|---|---|---|---|
| Cox | .193±.053 | .190±.061 | .206±.069 | .220±.071 | .233±.068 | .245±.072 |
| FG | .183±.051 | .186±.062 | .195±.067 | .212±.069 | .230±.070 | .234±.069 |
| BST | .169±.046 | .172±.044 | .177±.049 | .185±.046 | .185±.047 | .193±.048 |
| RF | .117±.045 | .151±.046 | .157±.043 | .169±.049 | .180±.051 | .185±.052 |
| LDR | **.111**±.035 | **.137**±.038 | **.142**±.036 | **.151**±.041 | **.165**±.044 | **.171**±.046 |

Table 9: Brier score for breast cancer of SEER.

|  | $\tau = 10$ | $\tau = 50$ | $\tau = 100$ | $\tau = 150$ | $\tau = 200$ | $\tau = 250$ | $\tau = 300$ |
|---|---|---|---|---|---|---|---|
| Cox | **.014**±.003 | **.106**±.006 | **.150**±.006 | .169±.006 | .177±.007 | **.180**±.006 | **.179**±.005 |
| FG | .016±.003 | .112±.011 | .156±.009 | .170±.006 | .177±.011 | .186±.013 | .189±.010 |
| BST | **.014**±.004 | .114±.008 | .154±.007 | **.168**±.004 | **.174**±.009 | .184±.009 | .184±.008 |
| RF | .015±.003 | **.106**±.007 | .151±.007 | .174±.008 | .182±.008 | .185±.008 | .187±.007 |
| LDR | .018±.003 | .107±.006 | .153±.006 | .173±.006 | .182±.007 | .186±.006 | .185±.006 |

FG and BST are comparable and all slightly outperform RF in terms of mean values. The C-indices of risk 2 for data 2 are also analogous to those of risk 1 as in panel (c) of Figure 1 except that LDR

Table 10: Brier score for other causes of SEER.

|  | $\tau = 10$ | $\tau = 50$ | $\tau = 100$ | $\tau = 150$ | $\tau = 200$ | $\tau = 250$ | $\tau = 300$ |
|---|---|---|---|---|---|---|---|
| Cox | **.008**±.003 | **.073**±.011 | **.141**±.010 | .195±.010 | **.204**±.010 | **.193**±.009 | **.178**±.007 |
| FG | **.008**±.003 | .076±.010 | .161±.013 | .241±.018 | .290±.029 | .302±.035 | .301±.040 |
| BST | **.008**±.003 | .074±.009 | .142±.011 | .201±.010 | .213±.016 | .203±.006 | .228±.018 |
| RF | **.008**±.003 | **.073**±.010 | .145±.011 | .200±.010 | .213±.009 | .207±.009 | .199±.008 |
| LDR | .009±.003 | .083±.008 | .148±.008 | **.193**±.008 | .205±.009 | .199±.008 | .194±.008 |

slightly underperforms RF in terms of mean values. But they both significally outperforms the other three approaches which completely fail.

(a) C-index of risk 2 for synthetic data 1.

(b) C-index of risk 2 for synthetic data 2.

Figure 5: Cause-specific C-indices of risk 2 for synthetic data 1 and 2.

Since we have random partitions in the analysis of DLBCL dataset, improvements of LDR can be underrated for the overlaps of boxplots across the five approaches in Figure 2. Therefore, we calculate the difference of C-indices between LDR and each of the other four benchmarks within each random partition, and report the mean and standard deviation in Table 11 where $\Delta_X$, $X \in \{Cox, FG, BST, RF\}$, denotes the C-index of LDR minus that of approach X. In terms of mean difference, LDR outperforms all the other benchmarks for all the three risks at any time evaluated except for BST under risk ABC.

Table 11: Difference of C-indices between LDR and other benchmarks.

|  | ABC | | | | GCB | | | | T3 | | | |
|---|---|---|---|---|---|---|---|---|---|---|---|---|
| year | $\Delta_{COX}$ | $\Delta_{FG}$ | $\Delta_{BST}$ | $\Delta_{RF}$ | $\Delta_{COX}$ | $\Delta_{FG}$ | $\Delta_{BST}$ | $\Delta_{RF}$ | $\Delta_{COX}$ | $\Delta_{FG}$ | $\Delta_{BST}$ | $\Delta_{RF}$ |
| 1 | .09±.08 | .03±.05 | .01±.06 | .06±.08 | .07±.09 | .06±.09 | .07±.06 | .16±.12 | .16±.15 | .11±.12 | .06±.05 | .10±.12 |
| 2 | .09±.06 | .03±.04 | .00±.07 | .04±.08 | .11±.08 | .10±.08 | .05±.06 | .17±.13 | .20±.17 | .10±.08 | .05±.05 | .03±.08 |
| 3 | .09±.05 | .04±.05 | -.01±.06 | .05±.06 | .12±.07 | .12±.06 | .05±.06 | .16±.09 | .20±.17 | .10±.09 | .05±.05 | .03±.08 |
| 4 | .09±.05 | .04±.05 | -.01±.06 | .05±.06 | .11±.07 | .12±.06 | .05±.06 | .15±.10 | .21±.15 | .11±.09 | .04±.05 | .02±.08 |
| 5 | .09±.05 | .04±.05 | -.01±.06 | .05±.06 | .12±.07 | .12±.06 | .05±.06 | .15±.09 | .21±.16 | .11±.08 | .04±.05 | .01±.08 |
| 6 | .09±.05 | .03±.05 | -.01±.06 | .04±.06 | .11±.07 | .12±.06 | .05±.06 | .16±.09 | .23±.14 | .11±.08 | .04±.05 | .02±.09 |