[Reviews · NeurIPS 2018]

Reviewer 1



-- Summary This paper describes a probabilistic model of competing risks and two algorithms for estimating its parameters: an MCMC algorithm and a scalable MAP estimation algorithm. The model has two appealing characteristics. First, it allows predictors to affect the hazard function non-linearly. Second, the non-linearity is formulated using latent "sub-events" that compete to determine when an observable event of interest will occur. This arguably makes the non-linearity more interpretable than approaches like random forests or multilayer perceptrons. Moreover, the number of sub-events is specified using a nonparameteric Bayesian model and so model complexity can adapt to the problem. -- Comments Competing risk models are often important in practice because events of interest rarely occur in isolation, and so I think the topic of this paper is important and will be interesting to the community. The model itself is clever, and introduces non-linearity through an intuitively appealing collection of latent sub-events. The experimental results show how posteriors over these latent variables can help to uncover subgroups of individuals who may have experienced the same event for different reasons. The number of sub-events affects the complexity of the model, and the authors propose an adaptive approach using Bayesian nonparametric priors to determine this piece of the model. The synthetic data results in Figure 1 do a nice job of showing how this prior adjusts the model complexity to the problem. It is also nice that the authors describe both an MCMC algorithm for inference and a MAP estimation algorithm as a scalable alternative. There are several ways that the paper could be strengthened. First, the concordance index is not necessarily a good indicator of predictive performance of event times. This is discussed, for example, in Section 3.4 of [1]. One alternative is the Brier score, which can measure the model's predictive ability as a function of time. Comparing the proposed model to baselines using something like the Brier score could provide more information about relative performance than the C-index. Second, it would be useful to understand whether the performance degrades when using the MCMC vs MAP estimation algorithms. This can be studied on smaller data sets where it's feasible to run the Gibbs sampler. Finally, there has been a lot of recent work on recurring event models (e.g. [2] and [3]). Although the competing risks problem is different, it may be helpful for readers to point out and discuss any connections to these recurring event models. -- Response to author feedback Thanks for including the new results using the Brier score. LDR appears to be competitive with the baselines. When there is more room, it would be useful to show the Brier score over time for each method rather than the average. It is also interesting that LDR slightly outperforms RF using C-index, but they switch when using Brier score. It would be useful to understand why this is happening, and what that says about the relative strengths and weaknesses of the models. -- References [1] Dynamic Prediction in Clinical Survival Analysis, Houwelingen and Putter [2] Isotonic Hawkes processes, Wang et al., 2016 [3] The neural Hawkes process, Mei and Eisner, 2017

Reviewer 2



This paper describes a novel approach to survival analysis, given multiple competing risks, which allows non-monotonic covariate effects. It provides a thorough theoretical foundation, then empirical results, based on both simulation data, and also some realworld data. The theorem appears correct (but did not check thoroughly), and the empirical results show that your approach appears effective. However, I wondered about the models you compared against, and the evaluation measure. It is great that you compared your system to several different models. It would be good to compare to other models that allow covariates to have different effects, at different times in the future: Deep Learning for Patient-Specific Kidney Graft Survival Analysis (https://arxiv.org/abs/1705.10245), DeepHit: A Deep Learning Approach to Survival Analysis with Competing Risks (http://medianetlab.ee.ucla.edu/papers/AAAI_2018_DeepHit.pdf), Learning patient-specific cancer survival distributions as a sequence of dependent regressors (http://papers.nips.cc/paper/4210-learning-patient-specific-cancer-survival-distributions-as-a-sequence-of-dependent-regressors ), A Multi-Task Learning Formulation for Survival Analysis (https://dl.acm.org/citation.cfm?id=2939857), etc. Also, concordance (and variants) are only one measure for evaluating survival models. It would be useful to understand how well their Lomax model is wrt calibration measures; see https://www.ncbi.nlm.nih.gov/pmc/articles/PMC3575184/ The nice example in li19-22 describes nonmonotonicness wrt feature values, where both high and low values can be problematic. Can this model also deal with situations where a single value of a variable can imply a high chance of dying soon, but low chance of dying at a later time? I had a hard time tracking sec3. Can you start with the diabetes example (li 136-144), and use that to explain the notation and the results? Would the user need to specify that type1 corresponds to S1, and type2 to S2? Or would this emerge from the data? Li197: I am puzzled why you exclude censored data from test set. Isn’t that the challenge of survival analysis? (I assume the training data included censored instances.) Li297: Should explain why the average is over just the uncensored instances Minor points: Li12: Lom(..) vs li 19 Lomax(...) Li128: “no longer a constant”. Is this wrt time? P4: wrt diabetes types: what is risk? Of event of death? Or hypoglycemic event? Or … P4: motivate why we need to deal with “countably infinite no. of atoms”? I thought the components would be known factors… is this to model ??? Li91: Here, in the embedded equation, is the \lambda = \sum_j \lambda_j? Eq 3, should the conditioned be { \lambda_j }, rather than \sum_j \lambda_j ? Fig1: shrinkalege_x000c_ ================== The authors did a nice job of addressing most of my comments--I especially like including Brier score (should note that li16 is only over uncensored data). Many measures do deal with some censored data -eg. c-index typically does include instances whose censored time is after the time of an uncensored instance, etc. Paper should explain why such methods don't apply here. My other remaining concerns are lack of comparisons with other non-proportional models, and the difficulty of tracking some parts of the article, I am willing to increase my assessment to 6.

Reviewer 3



The paper proposes a survival regression method for competing risks based on generalizing exponential distributions by using scale mixtures with gamma variables, giving Lomax distributions for the competing event times. This is further extended so that each competing event time can be divided into latent causes, basically using the same racing "trick" (taking minimum of the latent times) and gamma process. This allows some non-linearity in the covariate associations to the event times or hazards, since min-function is non-linear. A Gibbs sampling algorithm for full posterior inference and a MAP estimation algorithm are detailed in the Appendix. Experiments on synthetic data and real data are presented, showing good performance compared to some alternatives. The paper introduces an interesting parametric method for competing risk survival regression. Overall, it is well-written and clear. I'm not sure if it brings anything drastically new to survival analysis, but the methodology seems reasonable and possibly useful. The way to decompose a risk into latent causes seems especially interesting contribution. Main comments for possible improvements of the paper: (1) It's not really true that one can't tell how covariates affect the survival in Gaussian process models, although, of course, with flexible non-linear models and/or high-dimensional covariates it gets more difficult. For low-dim. example, see Joensuu et al. Risk of recurrence of gastrointestinal stromal tumour after surgery: an analysis of pooled population-based cohorts, Lancet Oncology, 2012. It would have been nice to have GP model(s) included in the results. (2) It's not very clear what is the benefit of the Lomax distribution for the sub-event times rather than any other parametric assumption (except over the exponential model, which has only a single parameter and leads to a constant baseline hazard model). It it mainly computational convenience in the availability of Gibbs sampling scheme? (How important is this when we now have probabilistic programming languages that are usually relatively efficient for (at least well-identifiable continuous parameter) regression models?) (3) The relation of the proposed approaches to previous literature could be more clearly delineated. Is is so that the Exponential racing is previously published but Lomax racing (and Lomax delegate racing) is original to this paper? No previous approaches to find latent causes of survival are discussed. (4) The interpretation of latent components is not easy/unambiguous as evident in the text as they are varyingly discussed as accounting for non-linearities or for subtypes of a competing risk. Minor comments: * Non-monotonic covariate effects don't necessarily break the PH assumption, but rather the linearity assumption? * I found the discussion around Equation 3 somewhat confusing; it tries to explain the data augmentation way of computing missing information/censored data, but didn't seem very clear. * Intuitively (so could be wrong...), with a model that takes min. over linear components, one would think that two equally weighted components would be reasonable in the Synthetic data 2, since the risk functions are symmetric? * typo: p.4, line 169: "Lamax" * Is the code available or will it be made available? Update after author feedback: This seems an interesting contribution for competing risks analysis and the author seem to acknowledge the main concerns of the reviewers in their response. (As a clarification to my point (4) above, if I remember correctly I meant something like that it's probably not really identifiable from data if there is "real" non-linearity of covariate effects or if the non-linearity arises from a combination of multiple linear latent causes. This might be important when interpreting the model inferences.)